# Ensuring Africa’s Food Security by 2050: The Role of Population Growth, Climate-Resilient Strategies, and Putative Pathways to Resilience

**DOI:** 10.3390/foods14020262

**Published:** 2025-01-15

**Authors:** Belay Simane, Thandi Kapwata, Natasha Naidoo, Guéladio Cissé, Caradee Y. Wright, Kiros Berhane

**Affiliations:** 1College of Development Studies, Center for Environment and Development Studies, Addis Ababa University, Sidist Kilo Campus, Addis Ababa 1176, Ethiopia; 2Environment and Health Research Unit, South African Medical Research Council, Johannesburg 2006, South Africa; thandi.kapwata@mrc.ac.za; 3Department of Environmental Health, Faculty of Health Sciences, University of Johannesburg, Auckland Park, P.O. Box 524, Johannesburg 2006, South Africa; 4Environment and Health Research Unit, South African Medical Research Council, Durban 4091, South Africa; natasha.naidoo@mrc.ac.za; 5Department of Epidemiology and Public Health, Swiss Tropical and Public Health Institute, CH-4002 Basel, Switzerland; gueladio.cisse@swisstph.ch; 6Ecosystem Health Sciences Unit, Faculty of Science, University of Basel, CH-4003 Basel, Switzerland; 7Environment and Health Research Unit, South African Medical Research Council, Pretoria 0084, South Africa; caradee.wright@mrc.ac.za; 8Department of Geography, Geoinformatics and Meteorology, University of Pretoria, Pretoria 0028, South Africa; 9Department of Biostatistics, Mailman School of Public Health, Columbia University, New York, NY 10032, USA

**Keywords:** food security, climate change, Africa, climate-resilient food systems, sustainable agriculture

## Abstract

Africa is grappling with severe food security challenges driven by population growth, climate change, land degradation, water scarcity, and socio-economic factors such as poverty and inequality. Climate variability and extreme weather events, including droughts, floods, and heatwaves, are intensifying food insecurity by reducing agricultural productivity, water availability, and livelihoods. This study examines the projected threats to food security in Africa, focusing on changes in temperature, precipitation patterns, and the frequency of extreme weather events. Using an Exponential Growth Model, we estimated the population from 2020 to 2050 across Africa’s five sub-regions. The analysis assumes a 5% reduction in crop yields for every degree of warming above historical levels, with a minimum requirement of 225 kg of cereals per person per year. Climate change is a critical factor in Africa’s food systems, with an average temperature increase of approximately +0.3 °C per decade. By 2050, the total food required to meet the 2100-kilocalorie per adult equivalent per day will rise to 558.7 million tons annually, up from 438.3 million tons in 2020. We conclude that Africa’s current food systems are unsustainable, lacking resilience to climate shocks and relying heavily on rain-fed agriculture with inadequate infrastructure and technology. We call for a transformation in food systems through policy reform, technological and structural changes, solutions to land degradation, and proven methods of increasing crop yields that take the needs of communities into account.

## 1. Introduction

Africa’s food security is increasingly at risk due to the complex effects of climate change, population growth, and dependence on rain-fed agriculture which makes the continent particularly vulnerable to climate disruptions. Africa remains one of the world’s most food-insecure regions, with millions of people experiencing chronic hunger and malnutrition. According to the Food and Agriculture Organization (FAO), as of 2023, approximately 282 million people in Africa are undernourished, representing over 20% of the continent’s population [1]. The prevalence of stunting among children under five years of age is especially alarming, with rates exceeding 30% in some areas, underscoring the severe nature of food insecurity across the continent [1].

Food security varies significantly across regions in Africa. West Africa and the Sahel face recurring food crises driven by drought and conflict, while East Africa, particularly Somalia, Kenya, and Ethiopia, suffers from prolonged droughts that have led to severe food shortages [1]. Southern Africa, although relatively more food secure, is increasingly impacted by climate change, with countries like Zimbabwe and Malawi experiencing periodic food crises [2]. North Africa, generally more food secure, faces challenges related to water scarcity and the effects of climate change on agriculture [3].

Despite these challenges, efforts are underway to improve food security across the continent. Initiatives such as the African Union’s Comprehensive Africa Agriculture Development Programme (CAADP) focus on enhancing agricultural productivity and resilience through investment in agriculture, infrastructure, and research. International organizations, governments, and NGOs are also working to provide emergency food aid, improve agricultural practices, and build resilience to climate change [4].

Given the ongoing threats posed by climate change to agricultural productivity, food security, and livelihoods across Africa, developing climate-resilient food systems is crucial. This requires a multifaceted approach, including crop diversification, improved water management, policy support, investment in research, and international cooperation. By implementing these strategies, Africa can protect its food security and ensure a sustainable future for its people [1,4]. This study emphasizes the urgency of building climate-resilient food systems to address the current and projected challenges to food security in Africa.

## 2. Materials and Methods

Relevant publications were sourced from Scopus, Web of Science, PubMed, ScienceDirect, and Google Scholar, focusing on data published between 2015 and 2023. We concentrated on the five sub-regions of Africa, as defined by the economic and political groupings used by the Intergovernmental Panel on Climate Change (IPCC) and the African Union (AU), as the unit of analysis for assessing food systems and formulating advisories (Figure 1) [5].

We acknowledge that climate change has already adversely impacted food security through decreased crop yields, degraded rangelands, diminished livestock and fishery production, reduced nutritional quality, and increased food prices (IPCC-AR6) [5]. According to the Intergovernmental Panel on Climate Change (IPCC), for many crops, a rise in temperature of approximately 1 °C is associated with a decrease in crop yields of about 5–10%, particularly in wheat and maize. For this analysis, we assumed a 5% reduction in crop productivity for every degree of warming above historical levels.

Population size and anticipated growth in Africa are crucial factors influencing future food consumption needs and the direction of agricultural development. Given the variability in projected population figures across sources and assumptions, we used the Exponential Growth Model to estimate regional population projections for 2050 [6].

The population projection formula was as follows [6]:P_t_ = P_0_ e^r × t^
where the following variables are used:

(P_t_) is the projected population at a future date;(P_0_) is the base year population;(e) is the natural logarithm base of 2.71828;(r) represents the growth rate of increase divided by 100;(t) represents the time period.

To assess food security, we used 225 kg of cereals per person per year as the minimum food requirement, equivalent to 2100 kilocalories per adult equivalent per day [7]. Food consumption, expressed in kcal/person/day, was the key variable used to measure and evaluate food security status in this study.

We also examined the health impacts of the food system, focusing on two anthropometric indices: stunting (low height for age) and wasting (low weight for height). These undernutrition indices are part of the 100 indicators used to track progress toward achieving the 17 Sustainable Development Goals (SDGs).

## 3. Results

### 3.1. State of Food Systems in Africa

#### Production Demand

Meeting the rising demand for sufficient and nutritious food in the face of the declining supply and quality of natural resources is a significant concern for Africa. This need has been triggered by the adverse threats related to climate change and population growth as well as the cultural and dietary changes associated with economic growth and development [8]. Meeting the challenge calls for yield increases and overall productivity gains in food and agricultural production in the context of socially, economically, and environmentally sustainable agriculture at a higher level. The productivity of most of Africa’s small-scale food producers is still below what could be achieved. This “yield gap” is usually a result of farmers being unable to access productivity-enhancing inputs and technologies and having insecure or inappropriate land access and tenure, as well as lacking knowledge and training opportunities (especially for women and young people entering rural labor markets) [9].

Africa is agro-ecologically and climatically diverse, with agro-environmental zones ranging from the tropical rainforests in West and Central Africa to the dry and arid zones of the Sahel. This diversity offers both opportunities and challenges. The productivity of agriculture in Africa remains low compared with other regions, a result of climate change and variability, land degradation caused by inappropriate farming practices, deforestation, mining activities, and desertification [10]. For instance, average yields for maize are about 2.5 times higher in Asia and South America and 6 times higher in North America [10]. As a wide variety of crops and commodities can be produced across Africa, a range of different solutions are required to overcome the varying bottlenecks that continue to limit the agricultural potential of the different countries and regions on the continent.

Africa has 60% of the world’s available arable land, and agriculture is the source of livelihood for 70% of the population [11]. Yet the continent generates only 10 percent of global agricultural output. This is because African agriculture is dominated by subsistence agriculture with low value addition and poor rural infrastructure. Africa’s agricultural systems are still largely dependent on rain with few technological inputs [12]. Only 5% of the cultivated land in Africa makes use of irrigation, with most of the farmers depending on rainfall.

The calculated trends in food requirements across Africa, both at the continental and regional levels, reveal a rising demand to support the growing population (Table 1). In 2020, a total of 438.3 million tons of food was needed to meet the standard of 2100 kilocalories per adult equivalent per day. By 2050, this requirement is projected to increase to 558.7 million tons annually, necessitating a 75% boost in food production compared to current levels.

The calculated produced food deficits in Africa and by region for the years 2020–2050 are presented in Table 2. The deficit for Africa during 2020 was 110.6 million metric tons and is expected to rise to 188.3 million metric tons by 2050 due to population growth if the current level of production intensity is not transformed (Table 2). While the deficit for West Africa will be the highest (99 million metric tons), Central Africa’s deficit will be the lowest (20.5 million metric tons). The total calculated produced food deficit for Africa is projected to reach 230.9 million metric tons by 2050. According to the African Union Development Agency (AUDA-NEPAD) report, Africa expended USD 43 billion on food imports in 2019, and this is projected to increase to USD 90 billion annually by 2030, pointing to two contrasting realities, namely existing opportunities for African agriculture and farmers and, on the other hand, growing unsustainable food supply dependence on foreign sources [13].

The food deficits produced are due to the high population pressure and low productivity of agriculture in Africa. Productivity remains low compared with other regions due to climate change and variability, land degradation caused by inappropriate farming practices, deforestation, mining activities, and desertification [10]. In addition, African rice yields are roughly half the levels seen in Asia, and North American rice yields are close to four times higher [10]. Climate change has already reduced food security through losses in crop yields, rangelands, livestock and fisheries, deterioration in food nutritional quality, access and distribution, and price spikes.

Table 3 highlights trends in undernourishment across various African regions alongside global patterns from 2005 to 2021. Globally, undernourishment rates decreased from 12.3% in 2005 to 7.6% in 2017 but subsequently rose, reaching 9.8% in 2021. In Africa, however, undernourishment remains markedly higher. Although Africa’s undernourishment rate fell from 20.7% in 2005 to a low of 15.8% in 2015, this decline reversed, climbing back to 20.2% by 2021. These patterns underline persistent food security challenges across the continent, with Central and Eastern Africa experiencing particularly high undernourishment rates. Addressing these regional disparities is crucial to advancing food security and improving nutrition in Africa.

After a long period of improvement between 2000 and 2013, malnourishment has increased substantially, and most of this deterioration occurred in 2019, 2020, and 2021 [14]. The number of undernourished people on the continent has risen by 47.9 million since 2014 and in 2021 stands at 278.0 million or nearly one-fifth of the population (Table 4). Of these, 15.6 million people are in Northern Africa and 234.7 million in sub-Saharan Africa.

This gradual deterioration of food security in Africa has been caused by various, often overlapping, drivers and pressures in the food system, including climate change-induced weather extremes, population growth, and economic declines due to COVID-19 [11]. In addition to climate change and population growth, the COVID-19 pandemic and Russia–Ukraine war have led to a dramatic economic downturn in Africa and contributed to the worsening food security situation by disrupting economic and livelihood activities [15,16]. In the short term, countries need to provide humanitarian assistance and implement effective food relief measures to improve food security and nutrition. Over the longer term, countries will need to prioritize investment in agriculture and related sectors, as well as in water, health, and education services to reduce vulnerabilities and build resilience to withstand shocks from climate change and conflicts, as well as economic downturns and slowdowns.

### 3.2. Population Projections and Food Security Assessment for 2050 in Africa

Africa’s growing population, with an estimated annual growth rate of 2.4%, is among the fastest and highest growth rates globally [17], placing an already constrained setting under significant pressure to meet rapidly increasing food security requirements. A rapidly growing population stresses ecosystems by increasing demand for food, energy, medicines, and water. Low productivity is linked to the limited availability and accessibility of agricultural inputs and the lack of access to climate-resilient technologies and machinery [18].

According to the 2019 United Nations Population Prospect, the population in Africa is growing at an exponential rate and becoming an increasing pressure on food security across Africa (Figure 2) [19]. This pressure is further exacerbated by climate change. The population growth rate in the continent is expected to decline from 2.54 in 2020 to 1.76 in 2050, yet the number of inhabitants could continue to increase significantly. By 2050, Africa could reach around 2.8 billion inhabitants, compared to 1.34 billion in 2020. The population density will grow from 45 in 2020 to 57 inhabitants per square kilometer in 2050 (Table 5).

Eastern and Western Africa are projected to experience the highest population growth from 445 (2020) to 974 (2050) million people and from 402 to 882 million people million people, respectively (Table 5). Central Africa’s population is also projected to increase by 231 million people, while the Northern (154 million) and Southern African (27 million) regions will experience the lowest growth rates by comparison. Rapid population growth, high fertility rates, low median age, increasing life expectancy, large households, and widespread poverty are the main features of Africa’s demographics [20].

The population density in Africa is projected to increase from 45 inhabitants per square kilometer in 2020 to 57 by 2050 (Table 6). East Africa and West Africa are the most densely populated regions on the continent, and this high density could pose significant challenges for developing sustainable food systems. Consequently, the demand for the region’s abundant land is expected to rise over the years, potentially leading to conflicts unless there is agricultural transformation [20].

From 2005 to 2020, the calculated population density rose from 31 to 43.6 persons per square kilometer. It is anticipated to reach 59.4 persons per square kilometer by 2050 (Table 7). This increasing demand for land across the continent reflects a growing population that relies heavily on land and its resources. Unfortunately, land productivity is low and has been declining, even as the majority of people depend on these resources. A rapidly growing population puts pressure on ecosystems by heightening the demand for food, energy, medicine, and water. This growth also distorts land use patterns and accelerates environmental degradation, resulting in soil erosion, deforestation, and loss of biodiversity [21].

### 3.3. Climate Change as a Driver of Food Systems in Africa

Climate change is increasingly jeopardizing food systems, particularly in Africa, where the impacts are becoming more pronounced. Rising temperatures, altered precipitation patterns, and extreme weather events are already reducing agricultural yields and disrupting food supply chains. Africa has seen a faster warming trend than the global average, with temperatures increasing by approximately +0.3 °C per decade between 1991 and 2021, compared to +0.2 °C per decade from 1961 to 1990 [21]. The IPCC reports that all six African sub-regions have experienced significant temperature increases, with North Africa showing the most notable rise at around 0.41 °C per decade from 1991 to 2021, more than double the region’s rate from 1961 to 1990 and nearly twice the global average [5].

Projections suggest that Southern Africa will become drier, while Eastern and Western Africa will face wetter conditions, heightening flood risks [12]. Sea levels along African coastlines, especially near the Red Sea and southwest Indian Ocean, are rising faster than the global average at nearly 4 mm/year [12]. This trend is expected to continue, increasing the risk of coastal flooding and saltwater intrusion, potentially affecting 108–116 million people by 2030 [12].

Rain-fed agriculture, which is critical for smallholder farmers, is particularly vulnerable to climate variability, leading to frequent crop losses, reduced yields, and heightened food insecurity. Since 1961, climate change has decreased Africa’s total productivity by 34%, with the greatest impacts in warmer regions like sub-Saharan Africa, where agricultural productivity is already low [23]. The IPCC (2023) predicts a 5–10% decline in crop productivity for every degree of warming above historical levels, further endangering the continent’s food security [22].

Projected climate warming under three climate change scenarios and the expected yield losses in % for each are presented for the different regions of Africa (Table 7). While the yield losses under the 2 °C level of global warming are 5.5% for West, Central, and East Africa, the expected losses in North Africa and South Africa will be 7.5% and 11.5%, respectively. On average, the expected yield losses in Africa will be 3.9%, 7.0%, and 12.2% under the 1.5 °C, 2 °C, and 3 °C levels of global warming. This will have a substantial adverse impact on food security in Africa and significantly increase the risk of hunger and is anticipated to coincide with low adaptive capacity as climate change intensifies other stressors [24,25]. This highlights the need to prioritize innovative measures for reducing vulnerabilities in African food systems [25,26].

However, there are many factors aside from climate change that influence future food systems and food security in Africa. The most relevant include the increasing population, demographic changes, social disruptions, natural resource degradation and technological and institutional issues in food production, processing, distribution, and markets.

### 3.4. Health Outcomes Due to Food Insecurity in Africa

Food insecurity and malnutrition are known to be major public health concerns across Africa. It is a challenge with huge social and economic costs, and the biggest risk factor for the African burden of disease [27]. Food insecurity continues to be one of the major causes of malnutrition in Africa. Nutrition during the first 1000 days (the first five years) of life is critical for the physical, cognitive, social, and emotional development of a child [27]. Irreversible effects on a child’s physical and mental health will occur if nutritional needs are not met during this period [27].

The anthropometric extent of the nutritious status of a child is grounded in height and weight ratios, and this is the most frequently used assessment method to check whether the child is properly nourished or undernourished [28]. Wasting and stunting are both associated with increased mortality, especially when both are present in the same child. Approximately 256.1 million people (20% of the total population) in Africa are undernourished; of these, 239.1 million live in sub-Saharan Africa. Within the sub-Saharan African region, the Southern African countries continue to have the lowest burden of undernourishment (5.3 million), while Eastern Africa has the highest burden in terms of numbers (133.1 million) [11].

Child growth is internationally recognized as an important indicator of nutritional status and health in populations [29]. Children under five years old are more vulnerable to malnutrition than any other age group, and their nutritional status is a sensitive indicator of their health status and nutrition [30]. The consequences of malnutrition are massive, pervasive, and often hidden. It stunts growth, erodes child development, reduces the amount of schooling children attain, and increases the likelihood of poverty in adulthood [1]. Stunting among children under five years old is associated with long-term effects on cognitive development, school achievement, economic productivity in adulthood, and maternal reproductive outcomes [31]. Although the percentage of stunting among children under five years old in Africa is gradually decreasing, it remains high at 30.7%, significantly above the global average, with the total number of stunted children still increasing (Table 8). Africa has 61.4 million stunted children, and while the prevalence fell between 2012 and 2020, the overall numbers rose over this period (although they fell in Eastern Africa and remained unchanged in Southern Africa). Between 2000 and 2020, the prevalence of stunting fell in nearly all regions in Africa. This is in part due to economic growth, with higher incomes leading to greater spending on health care and better diets.

The prevalence of wasting in Africa is 6.0%, which is lower than the global average of 6.7% (Figure 3). However, there is considerable variation across regions. While wasting in Southern Africa is particularly low (3.2%), West Africa has the highest level of wasting (6.9%). There are 12.1 million children wasted in Africa, 8 million of whom are in Eastern and Western Africa.

## 4. Discussion

Our study emphasizes the serious and expanding risk of food insecurity in Africa, which is fueled by the interrelated issues of socio-economic inequality, land degradation, climate change, and fast population expansion. The five sub-regions of Africa will have substantial increases in food demand, but agricultural production could decline as a result of temperature rise, altered precipitation patterns, and an increase in the frequency of extreme weather events. Scientific and technological advancements are essential to improving food security amidst the rapid African population growth projected to reach nearly 2.8 billion by 2050. Innovations in agriculture, such as precision farming technologies, enhance crop yields and resource efficiency. Climate-resilient crops help farmers adapt to challenges like drought and extreme weather. Sustainable practices, including agro-ecology and regenerative farming, protect the environment and soil health, while innovations in food storage and distribution reduce waste and improve access. Additionally, empowering smallholder farmers through mobile technologies and financing boosts productivity, contributing to long-term food security despite growing demand [4].

Food security in Africa faces mounting challenges due to rapid population growth, climate change, and resource constraints. To address these issues effectively, there is a pressing need for sustainable food system enhancement that integrates environmental, economic, and social dimensions into food production and distribution. Sustainable food systems in Africa aim to enhance agricultural productivity while reducing environmental impacts and promoting social equity. Sustainable food systems are vital for addressing food security challenges by promoting practices that ensure long-term resilience and resource efficiency [1]. Africa can build resilient food systems that can adapt to changing conditions and ensure a stable and nutritious food supply for its growing population by implementing strategies that address the production demand by the increasing population, adopting Climate-Smart Agriculture (CSA) and Nutrition-Sensitive Agriculture to reduce its health outcomes [32]. Sustainable food system enhancement represents a comprehensive approach to addressing food security challenges, promoting environmental sustainability, and fostering socio-economic development across the continent.

### 4.1. Addressing Production Demand

To meet the growing food demand in Africa, it is essential to address the existing productivity gaps in agriculture. These gaps are largely due to factors such as inadequate access to essential inputs and technologies, insecure land tenure, and insufficient training for farmers. Tackling these challenges through targeted investments in agricultural technology, land policy reforms, and enhanced training programs is crucial for boosting agricultural productivity [33].

One of the key strategies is to improve access to quality seeds, fertilizers, and modern farming technologies. This can be facilitated by developing better distribution networks, offering subsidies, and fostering partnerships with private sector stakeholders (IFPRI, 2019). Strengthening land tenure security through legal reforms and comprehensive land registration processes can also encourage farmers to invest more in their land, leading to significant productivity gains [34]. Additionally, enhancing market access by improving infrastructure and establishing robust market information systems can help farmers secure better prices for their produce, thereby motivating them to increase productivity [35].

Governments should invest in infrastructure, such as rural roads and storage facilities, to reduce logistical barriers. Subsidy programs can help smallholder farmers access essential inputs. Land policy reforms, including digital land registration, should be implemented to improve tenure security. Encouraging private sector partnerships through incentives like tax breaks will foster agribusiness development, while digital platforms for real-time market pricing can enhance market access.

### 4.2. Population and Food Security

The projected population growth in Africa is placing increasing pressure on food systems, posing significant challenges to sustainability. Rapid population expansion amplifies the demand for essential resources such as food, energy, medicines, and water, thereby straining ecosystems. This is particularly concerning in regions like East and West Africa, where high population growth rates could intensify competition for land and resources. Such pressures can lead to distortions in land use systems, contributing to environmental degradation through soil erosion, deforestation, and biodiversity loss [36].

To mitigate the impact of population growth on food systems in Africa, a multifaceted approach is required. Expanding access to family planning and reproductive health services is crucial for managing population growth [37]. Additionally, investing in education, particularly for women and girls, plays a vital role in slowing population growth and improving economic outcomes [38]. Creating more economic opportunities, especially in rural areas, is also essential for alleviating pressure on food systems [39]. Furthermore, the adoption of sustainable agricultural practices can help ensure that food production meets the rising demand without further degrading the environment [40]. Implementing policies that promote sustainable land use, agricultural innovation, and infrastructure development is equally important in addressing the challenges posed by population growth.

Expanding family planning services, particularly in rural areas, and implementing national education campaigns for girls can slow population growth. Governments should create rural employment opportunities through agribusiness and skill development programs. Sustainable land use practices should be incentivized through grants or loans for farmers.

### 4.3. Climate Change Adaptation

The impact of climate change on agriculture in Africa emphasizes the urgent need for adaptive measures to safeguard food security. Decreased crop yields in Africa are expected to significantly exacerbate food insecurity across the continent, particularly due to Africa’s limited adaptive capacity, as climate change exacerbates other existing stressors [24,25]. Consequently, it is imperative to prioritize innovative strategies aimed at reducing the vulnerabilities of African food systems [25,26].

It is also evident that environmental degradation beyond climate change, such as soil erosion, deforestation, water scarcity, biodiversity loss, and land degradation, presents significant challenges to food security in Africa, especially as the population continues to grow rapidly. Addressing these environmental issues is crucial for maintaining agricultural productivity, ensuring food availability, and improving overall human well-being. Integrated approaches that combine sustainable land use, technological innovations, effective governance, and community-based interventions will be essential for achieving long-term food security.

The diverse impacts of climate change across different regions of Africa, such as increasing aridity in Southern Africa and wetter conditions in Eastern and Western Africa, necessitate the development of region-specific adaptation strategies. To effectively mitigate these challenges, substantial investments in climate-resilient agricultural practices and infrastructure are essential. One key approach is the promotion of CSA practices, which are critical in addressing the adverse effects of climate change on agriculture [41]. CSA initiatives, including the adoption of drought-resistant crop varieties, the implementation of efficient irrigation systems, and the enhancement of soil management techniques, can significantly boost agricultural productivity while building resilience to climate variability [41].

Governments should support the adoption of Climate-Smart Agriculture (CSA) by providing subsidies for drought-resistant crops and efficient irrigation systems. Investments in climate-resilient infrastructure, such as water management systems, are crucial. Region-specific adaptation strategies should be developed to address local climate challenges and improve farming resilience.

### 4.4. Health Implications

The intricate relationship between food insecurity and poor health outcomes underscores the critical need to enhance food security as a means to improve public health in Africa. The long-term effects of malnutrition and stunting on both physical and cognitive development highlight the urgency of implementing comprehensive nutrition programs and interventions. To effectively address the health impacts of food insecurity, a multifaceted approach is required, one that tackles the underlying causes of food insecurity and integrates various sectors.

One key strategy is the implementation of Nutrition-Sensitive Agriculture, which involves integrating nutrition goals into agricultural policies and programs [42]. This approach ensures that increases in food safety during production are coupled with improvements in the nutritional quality of diets. Such integration is crucial for addressing both food security and malnutrition [42].

Another essential component is the expansion and improvement of food distribution systems and social safety nets [43]. These measures are vital for ensuring that vulnerable populations consistently have access to adequate and nutritious food, thus mitigating the immediate risks of food insecurity. In addition, providing comprehensive maternal and child health services plays a critical role in mitigating the adverse effects of food insecurity on health [44]. Services such as prenatal care, breastfeeding support, and micronutrient supplementation are essential in preventing malnutrition and ensuring healthy development.

Improving water, sanitation, and hygiene (WASH) infrastructure and practices is another crucial strategy. Enhanced WASH practices can significantly reduce the incidence of waterborne diseases, which are particularly prevalent in food-insecure regions where malnutrition often weakens immune systems. Evidence from several studies demonstrates the urgent need for improved WASH infrastructure for the improvement of health in low-income communities [45].

Individual governments should integrate Nutrition-Sensitive Agriculture into extension services based on their contexts to improve food security and health. Strengthening food distribution networks and expanding social safety nets will ensure consistent access to nutritious food. Additionally, improving water, sanitation, and hygiene (WASH) infrastructure is essential in food-insecure areas. These efforts can lead to reductions in malnutrition, disease burden, and mortality rates, ultimately fostering healthier and more resilient communities.

### 4.5. Gender Dimensions

The role of women in agriculture is indispensable to food security in Africa. By recognizing and addressing the unique challenges women face, gender-focused interventions can empower women, enhance their productivity, and improve household and community food security. Strengthening women’s access to land, resources, education, and financial services as well as promoting women’s leadership and decision-making are critical strategies for achieving sustainable agricultural growth and food security. Furthermore, overcoming cultural and societal barriers that limit women’s participation in agriculture will be essential for achieving long-term food security and gender equality in Africa. Empowering women in agriculture not only benefits women but also contributes to broader socio-economic development, poverty reduction, and food security for the entire population.

### 4.6. Limitations and Future Research

Accurate projections of population growth and food demand are crucial for planning future agricultural policies, food security strategies, and economic development in Africa. The limitations of data sources, combined with the variability in population and food demand projections, create significant challenges for policymakers and planners working to ensure food security in Africa.

This study focuses primarily on quantitative analyses and projections of food system resilience. Future research should consider the existing benchmarking of agricultural systems that demonstrate both high productivity and climate resilience. Collaborations with agricultural scientists could provide critical insights into production practices that enhance sustainability, offering a more holistic approach to addressing food security and climate adaptation challenges.

While the reliance on secondary data, such as FAOSTAT and UNDESA projections, is valuable for understanding trends in agriculture and population growth, it also has limitations and potential biases. These sources rely on estimates, and their accuracy can be affected by factors like incomplete data, assumptions made during projection models, and regional variability.

Another key limitation of this study is its focus on food quantity, with less emphasis on nutritional quality and its direct link to public health outcomes, particularly in vulnerable populations. Nutritional quality plays a critical role in determining health outcomes, and addressing both food availability and nutritional adequacy is essential for tackling malnutrition and promoting overall well-being. Future research should aim to incorporate a more comprehensive analysis of nutrition quality, exploring its intersection with food security and its broader implications for public health, especially in regions facing climate and food system challenges.

We did not include meat consumption in our analysis. Given the historical preference for animal products in many populations, we acknowledge that it is more relevant to the continent of Africa. While our focus on grains and cereals is intended to represent overall food demand, future research could explore the integration of both plant-based and animal-derived food systems to better understand their interdependence and potential for improving nutrition and food security. This approach could further enhance the scalability and sustainability of food system solutions across the region.

## 5. Conclusions

According to our analysis using the Exponential Growth Model, Africa’s present food systems will not be able to feed its growing population. The amount of cereal required by the growing population will rise dramatically from 438.3 million tons in 2020 to 558.7 million tons by 2050. Without targeted action, there is a significant risk of widespread food insecurity given these anticipated rising food demands and the continent’s reliance on rain-fed agriculture.

The current state of Africa’s food systems is unsustainable due to its lack of resilience to climate shocks, poor infrastructure, restricted access to technology, and reliance on erratic rainfall. A comprehensive strategy involving policy changes, investments in Climate-Smart Agriculture, technological advancements, and infrastructural upgrades is needed to address these issues. Crucially, endeavors to augment food production have to take into account the requirements and environments of nearby communities, emphasizing sustainable land administration and tested techniques to enhance crop yields.

In conclusion, there is a path toward a more resilient, sustainable, and equitable food system by 2050 with the correct policies and investments. Considering the rapidly changing climate, immediate action is required to stop further deterioration and guarantee that the continent’s expanding population has access to enough nourishing food.

## Figures and Tables

**Figure 1 foods-14-00262-f001:**
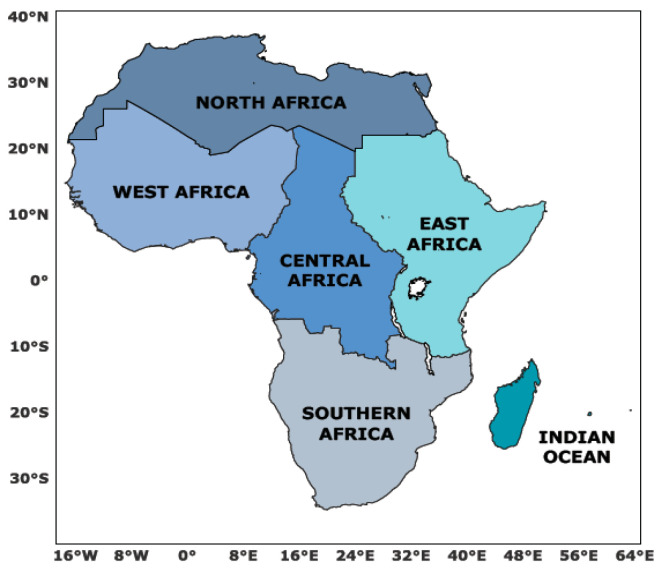
The five sub-regions of Africa.

**Figure 2 foods-14-00262-f002:**
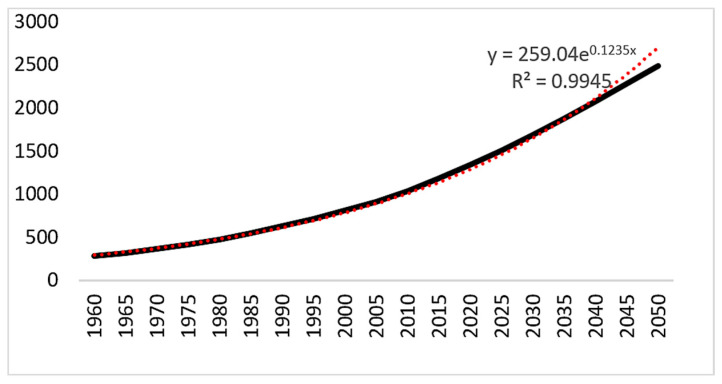
Historical and forecasted population growth in Africa. The solid black line represents the actual plotted data, while the red dotted line is a trend line extrapolated from the data points. The equation shown is derived from this trend line.

**Figure 3 foods-14-00262-f003:**
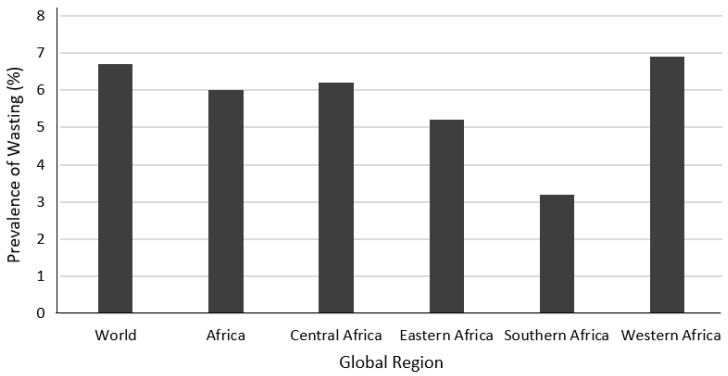
Prevalence of wasting among children under five years old in Africa by sub-region (2020).

**Table 1 foods-14-00262-t001:** Food requirement (million T/year). (Source: authors’ calculations based on the projected population growth in Africa).

	Regions in Africa					
Year	East (Million T/yr)	Central (Million T/yr)	North (Million T/yr)	South (Million T/yr)	West (Million T/yr)	Total (Million T/yr)
2020	100.2	40.4	55.4	151.9	90.4	438.3
2025	102.9	41.6	56.4	153.9	101.8	456.5
2030	105.5	42.9	57.4	155.8	113.9	475.6
2035	108.3	44.1	58.5	157.9	126.7	495.5
2040	111.1	45.5	59.6	159.9	140.1	516.0
2045	114.0	46.8	60.6	162.0	153.9	537.3
2050	116.9	48.2	61.7	164.0	167.9	558.7

**Table 2 foods-14-00262-t002:** Deficits of produced food due to population growth (mil tons/year).

	Regions in Africa					
Year	East (million T/yr)	Central (million T/yr)	North (million T/yr)	South (million T/yr)	West (million T/yr)	Total (million T/yr)
2020	16.2	12.7	29.0	31.0	21.6	110.6
2025	18.9	13.9	30.0	33.0	33.0	128.8
2030	21.6	15.2	31.1	34.9	45.1	147.8
2035	24.3	16.5	32.1	37.0	57.9	167.7
2040	27.1	17.8	33.2	39.0	71.3	188.3
2045	30.0	19.1	34.2	41.1	85.1	209.5
2050	32.9	20.5	35.3	43.1	99.1	230.9

Authors’ calculations based on FAOSTAT, https://www.fao.org/faostat/en/#data (accessed on 22 August 2023), and the projected population growth in Africa.

**Table 3 foods-14-00262-t003:** Prevalence of undernourishment in Africa by region, 2005–2021.

Global Region	Percentage of Undernourishment by Year								
	2005	2010	2015	2016	2017	2018	2019	2020	2021
World	12.3	8.6	8.0	7.8	7.6	7.7	8.0	9.3	9.8
Africa	20.7	16.5	15.8	16.3	16.4	17.0	17.4	19.6	20.2
North Africa	8.4	6.4	5.2	5.4	5.6	5.5	5.4	5.9	6.9
Eastern Africa	33.8	26.5	24.4	25.2	25.4	26.6	27.5	30.2	29.8
Central Africa	34.9	26.0	26.3	27.4	26.6	27.3	28.1	30.4	32.8
Southern Africa	4.9	5.8	7.4	7.4	7.5	7.4	7.9	9.1	9.2
Western Africa	12.2	9.9	10.1	10.1	10.0	10.6	10.4	13.2	13.9

**Table 4 foods-14-00262-t004:** Number of undernourished people in Africa by region.

Global Region	Year ^#^								
	2005	2010	2015	2016	2017	2018	2019	2020	2021
World	805.5	601.3	588.6	585.1	573.3	590.6	618.4	721.7	767.9
Africa	189.9	171.0	187.4	189.0	203.5	216.8	227.5	262.8	278.0
North Africa	15.6	13.0	11.6	12.2	13.1	13.1	13.1	14.6	17.4
Eastern Africa	99.8	89.9	95.2	100.9	104.6	112.3	119.3	134.4	136.4
Central Africa	39.1	34.2	40.6	43.6	43.6	46.2	48.9	54.7	60.7
South Africa	2.7	3.4	4.7	4.8	4.8	4.9	5.3	6.2	6.3
Western Africa	32.6	30.5	35.4	36.5	37.3	40.3	40.8	53.0	57.3

^#^ The numbers represented in the table are in millions.

**Table 5 foods-14-00262-t005:** Projected population growth in Africa by region.

	Regions in Africa					
Year	East	Central	North	South	West	Africa
2020	445.41	179.6	246.23	67.50	401.86	1340.604
2025	508.52	208.98	269.96	72.07	459.02	1518.55
2030	580.10	242.94	295.81	76.92	523.91	1719.687
2035	661.27	282.19	323.97	82.07	597.51	1946.999
2040	753.20	327.47	354.61	87.52	680.90	2203.702
2045	857.18	379.65	387.93	93.30	775.28	2493.343
2050	974.66	439.71	424.11	99.41	881.97	2819.867

Source: authors’ calculations based on 2019 United Nations Population Prospect population in Africa [19].

**Table 6 foods-14-00262-t006:** Projected population density in Africa by region.

	Regions in Africa ^#^					
Year	East	Central	North	South	West	Africa
2020	66.8	27.6	31.7	25.5	66.3	43.6
2025	68.5	28.5	32.3	25.8	74.6	45.9
2030	70.3	29.3	32.9	26.1	83.5	48.4
2035	72.2	30.2	33.5	26.5	92.9	51.0
2040	74.0	31.1	34.1	26.8	102.7	53.7
2045	76.0	32.0	34.7	27.2	112.9	56.5
2050	77.9	32.9	35.3	27.5	123.1	59.4

^#^ Numbers are representative of inhabitants per square kilometer. Source: authors’ calculations based on 2019 United Nations Population Prospect population in Africa [19].

**Table 7 foods-14-00262-t007:** Projected warming for 2050 compared to the 1994–2005 average above pre-industrial levels and calculated yield losses in Africa.

Region	Observed Warming Since the Mid-1970s (°C)	Projected Warming Under Different Level of Global Warming (°C)			Yield Reductions in % Under Different Level of Global Warming (%)		
		1.5 °C	2 °C	3 °C	1.5 °C	2 °C	3 °C
North Africa	1.2–2.4	0.9	1.5	2.6	4.5	7.5	13
West Africa	1.0–3.0	0.6	1.1	2.1	3	5.5	10.5
Central Africa	0.75–1.2	0.6	1.1	2.1	3	5.5	10.5
East Africa	0.7–1.0	0.6	1.1	2.1	3	5.5	10.5
South Africa	1.04–1.44	1.2	2.3	3.3	6	11.5	16.5
Africa		0.78	1.42	2.44	3.9	7.1	12.2

Source: authors’ elaboration based on IPCC, 2023 [22].

**Table 8 foods-14-00262-t008:** Prevalence of stunting among children under five years old in Africa.

Global Region	Stunting Among Children Under Five Years Old (%)				
	2000	2005	2010	2015	2020
World	33.1	30.7	27.7	24.4	22.0
Africa	44.5	39.1	35.9	32.8	30.7
North Africa	28.3	26.1	23.8	21.9	21.4
Eastern Africa	49.1	45.2	40.8	36.2	32.6
Central Africa	44.9	41.4	38.7	37.4	36.8
Southern Africa	29.1	28.5	25.0	23.5	23.3
Western Africa	39.9	38.5	36.1	33.4	30.9

Source: UNICEF, WHO, and World Bank [1].

## Data Availability

No new data were created or analyzed in this study. Data sharing is not applicable to this article.

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
