# Peer review of "Ensuring Africa’s Food Security by 2050: The Role of Population Growth, Climate-Resilient Strategies, and Putative Pathways to Resilience"

_foods, 2025, doi:10.3390/foods14020262_

Round 1
Reviewer 1 Report
Comments and Suggestions for Authors
Dear Author, • The authors focus on population growth and ignore the positive effects of knowledge progress and technological development. Also overlooked is the fact that the increase in the standard of living is often associated with a change in diet from plant-based to "carnivorous", which is associated with much higher, albeit indirect, consumption of plants. I suggest indication of posiitive effect of scientific and technology progress.
Reviewer 2 Report
Comments and Suggestions for Authors
Peer-Review Comments
Strengths
- Timely and Relevant Topic:
- The manuscript addresses critical issues of food security in Africa under the dual pressures of population growth and climate change, making it highly relevant to current global challenges.
- Methodological Clarity:
- The use of population growth models and food requirement projections up to 2050 is methodologically robust, providing a clear and systematic analysis.
- Policy and Practical Implications:
- Recommendations for climate-resilient agriculture and sustainable food systems are comprehensive and could guide policymakers and stakeholders.
Key Areas for Improvement
- Data Sources and Validation:
- The reliance on secondary data, such as FAOSTAT and UNDESA projections, requires further discussion on limitations and potential biases. Explicit mention of the variability or margin of error in these projections would strengthen the analysis.
- Lack of Focus on Regional Nuances:
- While the manuscript highlights regional differences in food security, it does not delve deeply into the sociopolitical and cultural factors driving these variations. For example, West Africa's projected food deficit could benefit from a discussion of governance and policy impacts.
- Gender Dimensions:
- The manuscript briefly mentions population growth but fails to discuss gender-related dynamics, such as the role of women in agriculture and family planning, which are crucial for food security and sustainable population growth.
- Environmental Degradation:
- While climate change is addressed, other environmental factors like soil degradation and biodiversity loss are underexplored. Integrating these would provide a more holistic view of food system challenges.
- Health and Nutrition:
- The manuscript emphasizes food quantity but could benefit from a stronger focus on nutrition quality and its link to public health outcomes, particularly in vulnerable populations.
Potential Errors or Oversights
- Projection Overlap: The manuscript uses multiple models (e.g., exponential growth) but does not discuss their comparability or provide cross-validation. This might confuse readers unfamiliar with statistical modeling.
- Visual Representation: Graphs and tables lack sufficient integration with the main text, reducing their interpretative value.
- Policy Implementation Challenges: Recommendations, while comprehensive, lack practical steps or case studies showcasing successful implementation of proposed strategies.
Specific Recommendations for Authors
- Provide detailed analysis on sociopolitical and cultural drivers behind regional differences in food security outcomes.
- Discuss limitations of data sources and variability in population and food demand projections.
- Expand on the role of women in agriculture and the significance of gender-focused interventions.
- Integrate more information on environmental degradation beyond climate change.
- Strengthen the link between food security and nutrition quality with concrete public health strategies.
Reviewer 3 Report
Comments and Suggestions for Authors
This manuscript deals with a relevant issues: food security and resilience in the food production systems.
This study examines the projected threats to food security across Africa´s five regions, focusing on population growth, agricultural productivity, changes in temperature, precipitation patterns, and the frequency of extreme weather events from 2020 to 2050.
The research issue, objective and method are clear and coherently stated.
The results are rich and soundly presented and discussed.
The the conclusions and implications are pertinent and supported by the results.
Comments and sugestions:
1 - Are there benchmarking agricultural production systems in the different regions that have high produtivity and better climate-resilience? If ok, what kind of production practices are they adopting? Into what extent this benchmarking practices can be suggested for and adapted in the different African regions?
2 - It seems pertinent to synthesize the content of Section 4 (Discussion) in a Table to facilitate the characterization and suggestions of the 4 dimensions discussed.
